# Exploring the In Vivo Existence Forms (23 Original Constituents and 147 Metabolites) of Astragali Radix Total Flavonoids and Their Distributions in Rats Using HPLC-DAD-ESI-IT-TOF-MS^n^

**DOI:** 10.3390/molecules25235560

**Published:** 2020-11-26

**Authors:** Li-Jia Liu, Hong-Fu Li, Feng Xu, Hong-Yan Wang, Yi-Fan Zhang, Guang-Xue Liu, Ming-Ying Shang, Xuan Wang, Shao-Qing Cai

**Affiliations:** State Key Laboratory of Natural and Biomimetic Drugs, School of Pharmaceutical Sciences, Peking University, No. 38 Xueyuan Road, Beijing 100191, China; lijialiu@bjmu.edu.cn (L.-J.L.); 1716383012@bjmu.edu.cn (H.-F.L.); wanghy@pku.org.cn (H.-Y.W.); zhangyf0911@pku.org.cn (Y.-F.Z.); guangxl@bjmu.edu.cn (G.-X.L.); myshang@bjmu.edu.cn (M.-Y. S.); xuanwang6818@bjmu.edu.cn (X.W.)

**Keywords:** Astragali Radix, flavonoids, metabolism, LC-MS, effective forms

## Abstract

Astragali Radix total flavonoids (ARTF) is one of the main bioactive components of Astragali Radix (AR), and has many pharmacological effects. However, its metabolism and effective forms remains unclear. The HPLC-DAD-ESI-IT-TOF-MS^n^ technique was used to screen and tentatively identify the in vivo original constituents and metabolites of ARTF and to clarify their distribution in rats after oral administration. In addition, modern chromatographic methods were used to isolate the main metabolites from rat urine and NMR spectroscopy was used to elucidate their structures. As a result, 170 compounds (23 original constituents and 147 metabolites) were tentatively identified as forms existing in vivo, 13 of which have the same pharmacological effect with ARTF. Among 170 compounds, three were newly detected original constituents in vivo and 89 were new metabolites of ARTF, from which 12 metabolites were regarded as new compounds. Nineteen original constituents and 65 metabolites were detected in 10 organs. Four metabolites were isolated and identified from rat urine, including a new compound (calycoisn-3’-*O*-glucuronide methyl ester), a firstly-isolated metabolite (astraisoflavan-7-*O*-glucoside-2’-*O*-glucuronide), and two known metabolites (daidzein-7-*O*-sulfate and calycosin-3’-*O*-glucuronide). The original constituents and metabolites existing in vivo may be material basis for ARTF efficacy, and these findings are helpful for further clarifying the effective forms of ARTF.

## 1. Introduction

Astragali Radix total flavonoids (ARTF) is one of the main bioactive components of Astragali Radix (AR, Huangqi in Chinese) [1]. Many studies have proven its cardiovascular protective effect, owing that it exhibited a protective effect on an ischemia-reperfusion model by effectively inhibiting the free radical spectrum [2,3], and exhibited vasorelaxant and endothelial-protective effect via the Akt/eNOS signaling pathway [4]. ARTF has an obvious protective effect on the inflammatory response in brain tissue of a natural aging rat by reducing the expression level of the downstream inflammatory factors [5]. ARTF also has immunostimulatory and anti-inflammatory effects via regulating MAPK (Mitogen-Activated Protein Kinase) and NF-κB signaling pathways [6,7]. ARTF has a protective effect against hepatic damage induced by paracetamol [8] or reperfusion [9] as well. In a word, ARTF has a wide range of pharmacological actions.

Up to now, besides studies on the pharmacological actions of AR, many investigations have been conducted in the field of phytochemistry. Over 70 flavonoid compounds have been isolated and identified from AR by modern chromatographic and spectroscopic methods [10,11,12], and 421 flavonoids have been detected and characterized from AR (the roots of *Astragalus membranaceus* (Fisch.) Bge. var. *mongholicus* (Bge.) Hsiao) by HPLC-MS (High Performance Liquid Chromatography-Mass Spectrometry) technology [13]. All these findings indicate that many kinds of constituents exist in ARTF. The metabolism of some high content flavonoid compounds in ARTF have been reported by our research group. Forty-one and 21 metabolites have been identified from the urine of rats after administration of calycosin-7-*O*-glucoside [14] and ononin [15], respectively. Twenty-six and 14 metabolites have been identified when calycosin [14] and formononetin [15] were incubated with rat liver S9. Relatively, the metabolic studies of isoflavan and pterocarpan are pretty few, and only our research group has done several before. Twenty-one and 20 metabolites of astrapterocarpan-3-*O*-glucoside, astraisoflavan-7-*O*-glucoside have been identified in rats, respectively [15]. Forty and 19 metabolites of astrapterocarpan [16] and astraisoflavan [15] have been detected when incubated with rat liver S9, respectively. And many other research groups also have identified some metabolites of these four main isoflavones (calycosin [17], calycosin-7-*O*-glucoside [18], formononetin [19,20], ononin [21]) in different test systems, such as human gut microbiota [18], zebrafish larvae [17], sheep [19], human liver microsome [20,21], and so on.

However, the forms existing in vivo (i.e., absorbed constituents and metabolites) of ARTF remains unclear. Our research group had done a study on ARTF metabolism before, and identified 127, 43, and 22 compounds in the urine, plasma, and feces, respectively [22]. But in that study, no glucuronides of ARTF were found, which should be main metabolites of flavonoid compounds; the distribution of ARTF was not investigated either. And the dose schedule of ARTF was once per day for seven days. As is well known, it usually needs a long dose period for traditional Chinese medicines to treat diseases in clinical practice. As for AR, it can be used for a long time [23]. Therefore, to better simulate the long dose period of AR and understand the existence forms in vivo of ARTF under long-term administration situation, this study was performed. The rats were orally administrated with ARTF twice a day for 31 days, and the samples of urine, plasma, feces, and organs were collected to identify original constituents and metabolites that existed in vivo, and to study the distribution of the existence forms in vivo. After that, the ARTF-containing urine was used to isolate some major metabolites.

## 2. Results and Discussion

To know which compounds exist in vivo, which is one of the prerequisites for the compounds to be effective forms, the compounds in the bio-samples including urine, plasma, feces, and organs of the rats after oral administration of ARTF were analyzed. In total, one hundred and seventy compounds (23 original constituents and 147 metabolites) were identified, among which 12 were regarded as new compounds (they are all metabolites) by retrieving information from the SciFinder database, three were newly detected original constituents, and 89 were new metabolites of ARTF. In 147 metabolites, nine were phase I and 138 were phase II metabolites. Among 138 phase II metabolites, ninety-two sulfates, twenty-six glucuronides, thirteen both sulfuric acid, and glucuronic acid conjugates and seven methyl conjugates were included, which indicated that sulfates were the most important existence forms of ARTF. To better understand the distribution of compounds, ten different organs of the rat were analyzed as well. Nineteen original constituents including six ones detected only in the organs (**F18**–**F23**) and 65 metabolites containing three ones detected only in the organs (**M145**–**M147**) were identified. Three, 5, 4, 3, 5, 17, 7, 7, 6 original constituents together with 3, 18, 5, 6, 22, 30, 46, 21, 6 metabolites were identified in the heart, liver, spleen, lungs, kidneys, stomach, small intestine, colon intestine, and thymus, respectively. However, no compounds were found in the brain. Four metabolites were isolated from the ARTF-containing urine and identified by NMR. To better understand potential effective forms among those compounds in vivo, we retrieved over 40 compounds which have specific structures from SciFinder, and 13 compounds including six original constituents, and seven metabolites were found to possess the same pharmacological effects as ARTF. Most of the phase Ⅱ metabolites were not found to possess bioactivity, perhaps owing that this kind of material was difficult to gain and to study.

### 2.1. Identification of Original Constituents and Metabolites of ARTF in Rats

#### 2.1.1. Identification of Original Constituents

The peaks which appeared at the same position in LC-MS chromatograms of both the administrated bio-samples and the ARTF but did not exist in the blank bio-samples were regarded as original constituents. By comparison of their extracted ion chromatograms (EICs) and base peak chromatograms (BPCs), twenty-three original constituents were identified (Table 1, **F1**–**F23**, and **F18**–**F23** only detected in the organs). They were composed of 15 flavones or isoflavones, namely calycosin and its isomers (**F1**–**F3**), calycosin-7-*O*-glucoside (**F4**), ononin (**F5**), formononetin (**F6**), isomers of pratensein (**F7**, **F8**), daidzein (**F11**), trihydroxyisoflavone/flavone (**F13**), isomer of odoratin (**F14**), dihydroxy dimethoxyisoflavone/flavone (**F16**), naringin (**F18**), pratensein glucoside and its isomer (**F22**, **F23**); four pterocarpans and isoflavanes, namely, astraisoflavane isomer (**F15**), astraptercarpan (**F17**), astrapterocarpan pentose glucoside (**F20**), 3,10-dihydroxy-9-methoxypterocarpan (**F21**); four dihydroisoflavones/flavones and chalcones, namely trihydroxy-dihydroisoflavone/flavone (**F9**), trihydroxy-tetrahydroisoflavone/flavone (**F10**), trihydroxychalcone (**F12**), dihydrocalycosin pentose glucoside (**F19**). Among the 23 original constituents, three were newly found original constituents of ARTF, namely **F3**, **F7**, and **F8**. Seventeen original constituents (**F1**–**F17**) were found in the urine (Figure 1); **F4**–**F6** (Appendix A) were detected in the plasma; **F1**, **F6**, **F12**, and **F17** (Appendix A) existed in the feces. The remaining six original constituents (**F18**–**F23**; Table 1) were detected only in the organs.

#### 2.1.2. Identification of Metabolites

The peaks only appearing in LC-MS chromatograms of ARTF-treated rat bio-samples, but not existing in either blank bio-samples or ARTF were regarded as metabolites. By comparing the EICs and BPCs of them, 147 peaks were assigned as metabolites (**M1**–**M147**; Table 2). **M145**–**M147** were only found in the organs. One hundred and six, 64, and 17 metabolites were identified in the urine (Figure 2), plasma (Appendix A), and feces (Appendix A), respectively. Among the 147 metabolites, 89 were new metabolites of ARTF, from which 12 were regarded as new compounds by searching information from SciFinder database (their MS information was shown in Appendix A and Appendix A). Eighty-nine new metabolites included the sulfates of the ring cleavage products of flavone, sulfates of oxidized, reduced, methylated astraisoflavan, and all the glucuronides as well as disulfates. And 12 potential new compounds were sulfates, disulfates, glucuronides, diglucuronides of tetrahydrocalycosin. By analyzing the structures of 147 metabolites, we speculated they were mainly derived from calycosin and its glycoside (maybe the sources of **M17**–**M72**, **M131**–**M139**), formononetin and its glycosides (maybe the sources of **M73**–**M78**, **M131**–**M139**), astrapterocarpan-3-*O*-glucoside (maybe the sources of **M79**–**M84**), astraisoflavan-7-*O*-glucisode (maybe the sources of **M85**–**M106**, **M147**) and many other low content constituents such as astrapterocarpan (maybe one of the sources of **M79**–**M84**), astraisoflavan (maybe one of the sources of **M85**–**M106**, **M147**), daidzein (maybe one source of **M107**–**M117**, **M131**–**M139**), genistein (maybe one source of **M118**–**M130**), and so on. Hence, they were speculated to be the main sources of existence forms of ARTF. These metabolites could be classified into 9 phase I metabolites and 138 phase II metabolites. A hundred and thirty-eight phase II metabolites consisted of 7 methyl conjugates, 92 sulfates, 26 glucuronides, and 13 both sulfuric acid and glucuronic acid conjugates, which indicated that sulfates were the most important existence forms of ARTF. The structural elucidation process of some representative metabolites was described as follows.

##### Identification of the Sulfates of the Ring Cleavage Products of Flavone (**M1**–**M16**)

A total of 16 compounds were assigned as metabolites originating from flavone, which underwent ring cleavage, then conjugation with sulfuric acid, and all of them were new metabolites of ARTF. **M1** showed [M − H]^−^ at *m*/*z* 200.99 and its molecular formula was predicted as C_7_H_6_O_5_S, and the fragment ion at *m*/*z* 121.03 was formed by the neutral loss of 79.95 Da (SO_3_) in its MS^2^ spectra, so it was determined as hydroxyl-benzaladehyde sulfate or other isomers. **M2** showed [M − H]^−^ at *m*/*z* 219.00 and its molecular formula was predicted as C_7_H_8_O_6_S, and the fragment ion generated by loss 79.95 Da could be detected at *m*/*z* 139.04, which was predicted as C_7_H_8_O_3_ and was one CH_2_ more than pyrogallol, so it was tentatively identified as methyl pyrogallol sulfate [24] or other isomers. **M3** and **M4** showed [M − H]^−^ at *m*/*z* 231.00, which indicated that their molecular formulae were C_8_H_8_O_6_S. In the MS^2^ spectra, fragment ions at *m*/*z* 151.04, 137.03 were formed by sequential losses of SO_3_ and CH_2_. According to literature, they were identified as hydroxy phenylacetic acid sulfate [25]. **M5** and **M6** showed [M − H]^−^ at *m*/*z* 243.00, which indicated that their molecular formulae were C_9_H_8_O_6_S, and according to the fragment ions at *m*/*z* 163.04, 119.05 and previous report, they were identified as hydroxycinnamic acid sulfate [26]. **M7** showed [M − H]^−^ at *m*/*z* 243.03 and was predicted as C_10_H_12_O_5_S. And according to the fragment ions at *m/z* 163.08, 148.05 in MS^2^ spectra, it was regarded as eugenol sulfate [27]. **M8** showed [M − H]^−^ at *m*/*z* 247.00 and was predicted as C_8_H_8_O_7_S, and yielded a fragment ion at *m/z* 167.03 by neutral loss of 79.95 Da (SO_3_), so it was determined as vanillic acid sulfate [28]. **M9** showed [M − H]^−^ at *m*/*z* 247.03, which indicated that their molecular formulae were C_9_H_12_O_6_S, and according to the fragment ions at *m/z* 167.07, 153.09, 137.05 and previous report, it was identified as homovanillyl alcohol sulfate [29]. **M10** showed [M − H]^−^ at *m*/*z* 273.00, which indicated that its molecular formula was C_10_H_10_O_7_S, and according to the fragment ions at *m*/*z* 178.03, 134.04, it was identified as ferulic acid sulfate [24]. **M11** and **M12** showed [M − H]^−^ at *m*/*z* 273.04, which indicated that their molecular formulae were C_11_H_14_O_6_S. And the fragment ions at *m/z* 193.09, 178.06, 163.03 were observed in MS^2^ spectra, which were 30.01 Da (OCH_2_) lager than that of the aglycon of **M7**. Therefore, they were tentatively determined as methoxyeugenol sulfate. **M16** showed [M − H]^−^ at *m*/*z* 303.02, which indicated that its molecular formula was C_11_H_12_O_8_S, and the fragment ions at *m/z* 223.06, 208.04, 164.05, 149.02, were similar to the fragment ions of 4-hydroxy-3,5-dimethoxycinnamic acid, so it was determined as 4-hydroxy-3,5-dimethoxycinnamic acid sulfate [30].

##### Identification of the Calycosin-Related Metabolites (**M17**–**M72, M146**)

**M17** showed [M − H]^−^ at *m*/*z* 283.06 and its molecular formula was predicted as C_16_H_12_O_5._ The fragment ions at *m*/*z* 269.04, 268.04, 195.04 were detected in MS^2^, which were like those of calycosin, so it was identified as calycosin isomer. **M18–M20** were identified as calycosin isomer owing that their predicted molecular formulae and fragment ions were like those of calycosin in the positive ion mode. **M21** showed [M + H]^+^ at 301.06 and its molecule formula was predicted as C_16_H_12_O_6,_ which had one more oxygen atom than that of calycosin, and the fragment ions at *m*/*z* 270.08, 197.08 were similar to those of calycosin in positive ion mode, so it was determined as hydroxycalycosin. The fragment ion of [aglycon−H]^−^ at *m*/*z* 283.06 could be detected in the MS^2^ spectra of **M25–M34**, which was predicted as C_16_H_12_O_5_, and other fragment ions were like those of calycosin, so they were determined as calycosin metabolites. **M25**–**M30** were predicted as C_16_H_12_O_8_S according to [M − H]^−^ at *m/z* 363.02, and the [aglycon − H]^−^ formed by the neutral loss of 79.95 Da (SO_3_) was detected. Therefore, they were determined as calycosin sulfate and its isomers. In the same way, **M30** was determined as calycosin-7,3’-*O*-disulfate. **M31** and **M146** showed [M − H]^−^ at *m*/*z* 459.09 and their molecular formulae were predicted as C_22_H_20_O_11,_ and the fragment ion of [aglycon − H]^−^ were formed by the neutral loss of 176.03 Da (C_6_H_8_O_6_) in the MS^2^ spectra, so they were determined as calycosin glucuronide. There are two positions (C-7-OH and C-3’-OH) in calycosin that could be linked with glucuronic acid, and C-3’-OH was the major position according to a previous study [31]. In addition, **M31** was the main metabolite, hence it was determined as calycosin-3’-*O*-glucuronide, and **M146** was determined as calycosin-7-*O*-glucuronide.

In the MS^2^ spectra of **M35**–**M39,** the fragment ion of [aglycon − H]^−^ at *m*/*z* 299.05 can be detected, which were predicted as C_16_H_11_O_6_ and was the same as **M21**, so they were tentatively determined as metabolites of hydroxycalycosin. And according to molecule formulae and characteristic neutral losses, they were tentatively determined as hydroxycalycosin sulfate or glucuronide, respectively.

**M42** showed [M − H]^−^ at *m*/*z* 621.15 and its molecular formula was predicted as C_28_H_30_O_16_. The fragment ion at *m*/*z* 283.06 predicted as C_16_H_11_O_5_ was detected in MS^2^ which was generated by sequential loss of 162.05 Da (C_6_H_10_O_5_), and 176.03 Da (C_6_H_8_O_6_). Since calycosin-7-*O*-glucoside was the main constituent of ARTF, so it was determined as calycosin-7-*O*-glucoside-3’-*O*-glucuronide.

The fragment ion of [aglycon − H]^−^ at *m*/*z* 285.08 could be detected in the MS^2^ spectra of **M43**–**M46,** which was predicted as C_16_H_13_O_5_ and had 2H (2.01 Da) more than that of calycosin, so they were tentatively regarded as metabolites of dihydrocalycosin. The fragment ion of [aglycon − H]^−^ at *m*/*z* 287.09 could be detected in the MS^2^ spectra of **M54**–**M63**, which were predicted as C_16_H_15_O_5_ and had 4H more than that of calycosin, so they were tentatively regarded as metabolites of tetrahydrocalycosin. In the MS^2^ spectra of **M64**–**M69,** the fragment ion of [aglycon − H]^−^ at *m*/*z* 303.08 could be detected, which were predicted as C_16_H_15_O_6_ and had one more oxygen atom than that of tetrahydrocalycosin, so they were tentatively regarded as hydroxy tetrahydrocalycosin metabolites.

##### Identification of the Formononetin-Related Metabolites (**M73**–**M78**)

**M73** showed [M − H]^−^ at *m*/*z* 347.02 and its molecular formula was predicted as C_16_H_12_O_7_S. The fragment ion of *m*/*z* 267.07 in the MS^2^ spectra was formed by the neutral loss of 79.95 (SO_3_), which was predicted as C_16_H_11_O_4_, and its fragment ion *m*/*z* 252.03 was like that of formononetin. Because only C-7-OH could be sulfated, so **M73** was determined as formononetin-7-*O*-sulfate. In the same way, **M74** was determined as formononetin-7-*O*-glucuronide. **M75**–**M77** showed [M − H]^−^ at *m*/*z* 351.05 and their molecular formulae were predicted as C_16_H_16_O_7_S. The fragment ions at *m*/*z* 271.09 formed by neutral loss of 79.95 Da (SO_3_) was determined as C_16_H_15_O_4_, which had 4H more than that of formononetin, so they were tentatively determined as tetrahydroformononetin sulfate.

##### Identification of the Astrapterocarpan-Related Metabolites **(M79**–**M84)**

**M79** showed [M − H]^−^ at *m*/*z* 379.05 and its molecular formula was predicted as C_17_H_16_O_8_S. The fragment ions *m*/*z* 299.08 formed by a natural loss of 176.03 Da (C_6_H_8_O_6_) was predicted as C_17_H_15_O_5_, which was the same to that of astrapterocarpan. Owing that only C-3-OH of astrapterocarpan could be linked to sulfuric acid, it was determined as astrapterocarpan-3-*O*-sulfate. In the same way, **M80** was determined as astrapterocarpan-3-*O*-glucuronide.

**M82**–**M84** showed [M − H]^−^ at *m*/*z* 491.12 and their molecular formulae were predicted as C_23_H_24_O_12_. The fragment ions at *m*/*z* 315.09 generated by neutral loss of 176.03 Da (C_6_H_8_O_6_) was predicted as C_17_H_15_O_6_, and had one more O than that of astrapterocarpan, so they were tentatively determined as hydroxyastrapterocarpan glucuronide.

##### Identification of the Astraisoflavan-Related Metabolites **(M85**–**M106, M147)**

**M85** showed [M + H]^+^ at *m*/*z* 303.13 and its molecular formula was predicted as C_17_H_18_O_5._ The fragment ions at *m*/*z* 167.10, 149.09, and 125.07 could be detected in MS^2^ spectra and were like those of astraisoflavan in positive ion mode. So, **M85** was determined as astraisoflavan isomer.

**M102**–**M103** showed [M − H]^−^ at *m*/*z* 477.14 and their molecular formulae were predicted as C_23_H_26_O_11_. The fragment ions at *m*/*z* 301.10 in MS^2^ spectra formed by a neutral loss of 176.03 Da (C_6_H_8_O_6_) was the same to that of astraisoflavan. Therefore, they were determined as astraisoflavan glucuronide. Because there are only two glucuronidation sites (C-7-OH, C-2’-OH) in astraisoflavan, and a larger CLogP value means a lower polarity and a larger retention time in reversed phase HPLC, **M102** (CLogP = 3.6083, t_R_ = 58.702 min) was determined as astraisoflavan-7-*O*-glucuronide and **M103** (CLogP = 3.2673, t_R_ = 57.802 min) was determined as astraisoflavan-2’-*O*-glucuronide.

##### Identification of the Daidzein-Related Metabolites (**M107**–**M117**)

**M107** and **M108** showed [M − H]^−^ at *m*/*z* 333.00 and their molecular formulae were predicted as C_15_H_10_O_7_S. The fragment ion at *m/z* 253.05 in MS^2^ spectra formed by the neutral loss of 79.95 Da (SO_3_), and it was predicted as daidzein owing that the fragment was predicted as C_15_H_9_O_4_ and *m*/*z* 225.05, 197.06, 135.01 were detected in MS^3^ spectra. Because there are two sulfation sites (C-7-OH, C-4’-OH) in daidzein, and a larger CLogP value means a lower polarity and a larger retention time in reversed phase HPLC, **M107** (CLogP = 0.4985, t_R_ = 41.557 min) was determined as daidzein-4’-*O*-sulfate and **M108** (CLogP = 0.3050, t_R_ = 28.925 min) was determined as daidzein-7-*O*-sulfate.

**M112** showed [M − H]^−^ at *m*/*z* 335.02 and its molecular formula was predicted as C_15_H_12_O_7_S. In MS^2^ spectra, the fragment ion at *m*/*z* 255.06 was formed by the neutral loss of 79.95 Da (SO_3_), which was predicted as C_15_H_11_O_4_ and had 2H more than that of daidzein. Therefore, it was tentatively determined as dihydrodaidzein sulfate.

**M113**–**M116** showed [M − H]^−^ at *m*/*z* 337.04 and their molecular formulae were predicted as C_15_H_14_O_7_S. The fragment ion at *m*/*z* 257.08 in MS^2^ spectra formed by the neutral loss of 79.95 Da (SO_3_), which was predicted as C_15_H_13_O_4_ and had 4H more than that of daidzein. Therefore, they were tentatively determined as tetrahydrodaidzein sulfate.

##### Identification of the Genistein-Related Metabolites (**M118**–**M130**)

**M118** showed [M + H]^+^ at *m*/*z* 271.06 and its molecule formula was predicted as C_15_H_10_O_5_. The fragment ions at *m*/*z* 253.01, 225.06, 215.07 which were like those of genistein in reported literature [32], so it was determined as genistein.

**M123**–**M125** showed [M − H]^−^ at *m*/*z* 351.02 and their molecular formulae were predicted as C_15_H_12_O_8_S. The fragment ion at *m*/*z* 271.06 was formed by a neutral loss 79.95 Da (SO_3_), which was predicted as C_15_H_11_O_5_ and had 2H more than that of genistein, so they were tentatively determined as dihydrogenistein sulfate.

**M126** showed [M − H]^−^ at *m*/*z* 273.07 and its molecular formula was predicted as C_15_H_14_O_5_, which was 4H more than that of genistein, so it was tentatively determined as tetrahydrogenistein. **M130** showed [M − H]^−^ at *m*/*z* 515.09 and its molecules formula was predicted as C_21_H_24_O_13_S. The fragment ion at *m*/*z* 273.07 which was predicted as C_15_H_13_O_5_ formed by a neutral loss of 162.05 Da (C_6_H_10_O_5_) and 79.95 Da (SO_3_), so it was tentatively determined as tetrahydrogenistin sulfate.

##### Identification of the Equol-Related Metabolites (**M131**–**M139**)

In the MS^2^ spectra of **M131**–**M138,** the fragment ion of [aglycon − H]^−^ at *m*/*z* 241.09 could be detected, which were predicted as C_15_H_13_O_3_ and its fragment ions of *m*/*z* 135.05, 121.04, 119.06 were like those of equol [33], so they were regarded as equol metabolites. **M136**–**M138** showed [M − H]^−^ at *m*/*z* 497.07 and their molecules formulae were predicted as C_21_H_22_O_12_S, and the [aglycon − H]^−^ formed by the neutral loss of 176.03 Da (C_6_H_8_O_6_), 79.97 Da (SO_3_). Therefore, they were determined as equol glucuronide sulfate. **M139** showed [M − H]^−^ at *m*/*z* 323.06 and its molecular formula was predicted as C_15_H_16_O_6_S. The fragment ion at *m*/*z* 243.10 formed by a neutral loss of 79.95 Da (SO_3_), which was predicted as C_15_H_15_O_3_ and 2H (2.01Da) more than that of equol. Hence, it was tentatively determined as dihydroequol sulfate.

##### Identification of the other Metabolites (**M140**–**M145**)

**M140** showed [M + H]^+^ at *m*/*z* 303.09 and its molecular formula was predicted as C_16_H_14_O_6_, which had 2H more than that of pratensein. Therefore, it was tentatively determined as dihydropratensein. **M141** and **M145** showed [M − H]^−^ at *m*/*z* 555.05 and their molecular formulae were predicted as C_22_H_20_O_15_S. The fragment ions at *m*/*z* 299.05 predicted as C_16_H_11_O_6_ was formed by a neutral loss of 176.03 Da (C_6_H_8_O_6_) and 79.95 Da (SO_3_). Therefore, it was tentatively determined as pratensein glucuronide sulfate.

### 2.2. Distribution of Original Constituents and Metabolites of ARTF in Rats Organs

#### 2.2.1. Distribution of Original Constituents

Nineteen original constituents were detected in the organs, with zero in brain, three in heart, five in liver, four in spleen, three in lung, five in kidney, seventeen in stomach, seven in small intestine, seven in colon intestine, and six in thymus, respectively (Appendix A, Appendix A). Six (**F18**–**F23**) were only detected in the organs and were not detected in the urine, plasma, and feces. Calycosin (**F1**), formononetin (**F6**), daidzein (**F11**), and naringin (**F18**) were widely distributed, which could be detected in seven and even more organs, and these compounds may be important material basis for the efficacy of ARTF.

#### 2.2.2. Distribution of Metabolites

Sixty-five metabolites were identified in the organs, of which three metabolites (**M145**–**M147**) were only detected in the organs, and 0, 3, 18, 5, 6, 22, 30, 46, 21, and 6 were identified in the brain, heart, liver, spleen, lung, kidney, stomach, small intestine, colon, and thymus (Appendix A, Appendix A), respectively. Twelve metabolites containing seven sulfates and five glucuronides were distributed widely in five and even more tissues. Seven sulfates were calycosin sulfate (**M26**), tetrahydrocalycosin sulfate (**M57**, **M58**), hydroxyastraisoflavan sulfate (**M96**), daidzein-4’-*O*-sulfate (**M107**), equol sulfate (**M132**, **M133**), respectively. Five glucuronides consisted of calycosin-3’-*O*-glucuronide (**M32**), dimethoxy hydroxytetrahydrocalycosin glucuronide (**M72**), astrapterocarpan-3-*O*-glucuronide (**M80**), astraisoflavan-2’-*O*-glucuronide (**M103**), and astraisoflavan-7-*O*-glucoside-2’-*O*-glucuronide (**M106**). These widely distributed metabolites may play an important role in the efficacies of ARTF.

### 2.3. Identification of Metabolites Isolated from Rat Urine

**MI-1** (**M108**) was obtained as a white powder and assigned a molecular formula of C_15_H_10_O_7_S based on its HR-ESI-MS mass spectrum, which showed a quasi-molecular ion peak [M − H]^−^ at *m*/*z* 333.0076 (calcd. for C_15_H_10_O_7_S 333.0069). The main fragment ion was *m*/*z* 253.0505 [M − SO_3_ − H]^−^ in MS^2^ spectra, so it was regarded as a sulfate. **MI-1** (**M108**):^13^C-NMR (DMSO-d_6_, 100MHz) ppm: 153.6 (C-2), 122.5 (C-3), 175.0 (C-4), 129.7 (C-5), 118.0 (C-6), 158.1 (C-7), 107.1 (C-8), 156.6 (C-9), 119.0 (C-10), 123.7 (C-1’), 130.2 (C-2’, C-6’), 115.08 (C-3’, C-5’), 157.3 (C-4’), which were in consistent with daidzein [34]. ^1^H-NMR spectra of **MI-1** (**M108**) (DMSO-d_6_, 400MHz) ppm: 8.38 (1H, s, H-2), 8.03 (1H, d, *J* = 8.8Hz, H-5), 7.43 (1H, d, *J* = 2.1Hz, H-8), 7.40 (2H, d, *J* = 8.3Hz, H-2’ and H-6’), 7.25 (1H, dd, *J* = 8.8Hz, 2.1Hz, H-6), 6.81 (2H, d, *J* = 8.3Hz, H-3’ and H-5’), which were like those of daidzein-7-*O*-sulfate reported in literature [35].Given all of this, **MI-1** was determined as daidzein-7-*O*-sulfate. Its structure and NMR spectroscopy were shown in Figure 3a and Appendix A, respectively.

**MI-2** (**M32**) was obtained as a white powder and assigned a molecular formula of C_22_H_20_O_11_ based on its HR-ESI-MS mass spectrum, which showed a quasi-molecular ion peak [M − H]^−^ t *m*/*z* 459.0943 (calcd. for C_22_H_20_O_11_ 459.0935). The main fragment ion was *m/z* 283.0609 [M − C_6_H_8_O_6_ − H]^−^ in MS^2^ spectra, so it was regarded as a glucuronide. **MI-2** (**M32**): ^1^H-NMR spectra (DMSO-d_6_, 400MHz) ppm: 8.31 (1H, s, H-2), 7.97 (1H, d, *J* = 8.8Hz, H-5), 6.94 (1H, dd, *J* = 8.8Hz, 2.2Hz, H-6), 6.87 (1H, d, *J* = 2.2Hz, H-8), 7.29 (1H, d, *J* = 2.0Hz, H-2’), 7.04 (1H, d, *J* = 8.5Hz, H-5’), 7.23 (1H, dd, *J* = 8.5Hz, 2.0Hz, H-6’), 3.79 (3H, s, C-4’-OC*H*_3_), 10.88 (1H, s, C-7-O*H*). ^13^C-NMR (DMSO-d_6_, 100MHz) ppm: 153.5 (C-2), 123.3 (C-3), 174.7 (C-4), 127.4 (C-5), 115.4 (C-6), 162.7 (C-7), 102.3 (C-8), 157.5 (C-9), 116.4 (C-10), 124.5 (C-1’), 116.7 (C-2’), 145.7 (C-3’), 149.2 (C-4’), 112.6 (C-5’), 123.3 (C-6’), 55.9 (C-4’-O*C*H_3_), which were similar to ^13^C-NMR of calycosin reported in literature [36]. The characteristic signals of six carbons in glucuronide were 100.2 (C-1’’), 73.1 (C-2’’), 76.3 (C-3’’), 71.6 (C-4’’), 75.6 (C-5’’), 170.2 (C-6’’), and all were consisted with calycosin-3’-*O*-glucuronide [31]. Based on the above analysis, **MI-2** (**M32**) was determined as calycosin-3’-*O*-glucuronide. Its structure and NMR spectroscopy were shown in Figure 3b and Appendix A, respectively.

**MI-3** was obtained as a white powder and assigned a molecular formula of C_23_H_22_O_11_ based on its HR-ESI-MS mass spectrum, which showed a quasi-molecular ion peak [M − H]^−^ at *m*/*z* 473.1113 (calcd. for C_23_H_22_O_11_ 473.1084). The main fragment ion was *m*/*z* 283.0601 [M − C_7_H_10_O_6_ − H]^−^ in MS^2^ spectra, so it was predicted as a glucuronide methyl ester. ^1^H-NMR (DMSO-d_6_, 400MHz) ppm: 8.31 (1H, s, H-2), 7.97 (1H, d, *J* = 8.7Hz, H-5), 6.94 (1H, dd, *J* = 8.7Hz, 2.2Hz, H-6), 6.87 (1H, d, *J* = 2.2Hz, H-8), 7.29 (1H, d, *J* = 2.0Hz, H-2’), 7.05 (1H, d, *J* = 8.5Hz, H-5’), 7.23 (1H, dd, *J* = 8.5Hz, 2.0Hz, H-6’), 3.79 (3H, s, C-4’-OC*H*_3_), 10.77 (1H, s, C-7-OH), 3.62 (3H, s, C-6’’-OC*H*_3_). ^13^C-NMR (DMSO-d_6_, 100MHZ) ppm: 153.4 (C-2), 123.2 (C-3), 174.5 (C-4), 127.3 (C-5), 115.3 (C-6), 162.8 (C-7), 102.1 (C-8), 157.4 (C-9), 116.2 (C-10), 124.5 (C-1’), 116.6 (C-2’), 145.5 (C-3’), 149.0 (C-4’), 112.4 (C-5’), 123.0 (C-6’), 55.8 (C-4’-O*C*H_3_), were carbon signals of calycosin [31], 99.9 (C-1’’), 73.0 (C-2’’), 75.8 (C-3’’), 71.4 (C-4’’), 75.2 (C-5’’), 169.2 (C-6’’) were carbon symbols of glucuronide, which were similar to those of calycosin-3’-*O*-glucuronide [31]. Compared with that, an additional methoxy carbon signal at δ52.0 was observed, and H signal of this methoxy at δ3.62 (3H, s, C-6’’-OC*H*_3_) was in correlated with carbonyl carbon signal of glucuronide at δ169.2 (C-6’’) in HMBC spectra, indicating that the methoxy was linked to the carbonyl of glucuronide. According to all above analysis, **MI-3** was identified as calycoisn-3’-*O*-glucuronide methyl ester (Figure 3c). It was a new compound. And its NMR spectroscopy was shown in Appendix A. But unfortunately, **MI-3** could not be detected in the bio-samples using the HPLC-DAD-ESI-IT-TOF-MS^n^ technique. Therefore, **MI-3** was maybe produced during the isolation process.

**MI-4** (**M106**) was obtained as a faint yellow powder and assigned a molecular formula of C_29_H_36_O_16_ based on its HR-ESI-MS mass spectrum, which showed a quasi-molecular ion peak [M − H]^−^ at *m*/*z* 639.1959 (calcd. for C_22_H_20_O_11_ 639.1925). The main fragment ion was *m*/*z* 463.1650 [M − C_6_H_10_O_5_ − H]^−^, 301.1066 [M − C_6_H_10_O_5_ − C_6_H_8_O_8_ − H]^−^ in MS^2^ spectra, so it was regarded as glucoside and glucuronide. ^1^H-NMR (DMSO-d_6_, 400MHz) ppm: 3.86 (1H, t, H-2a), 4.23 (1H, d, *J* = 8.0Hz, H-2b), 3.59 (1H, m, H-3), 2.74 (1H, m, H-4a), 2.84 (1H, m, H-4b), 6.98 (1H, d, *J* = 8.4Hz, H-5), 6.54 (1H, dd, *J* = 8.4 Hz, 2.4Hz, H-6), 6.48 (1H, d, *J* = 2.4Hz, H-8), 6.81 (1H, d, *J* = 8.8Hz, H-5’), 6.92 (1H, d, *J* = 8.8Hz, H-6’), 3.72 (3H, s, C-3’-OC*H*_3_), 3.77 (3H, s, C-4’-OC*H*_3_). ^13^C-NMR (DMSO-d_6_, 100MHZ) ppm: 69.8 (C-2), 30.0 (C-3), 30.9 (C-4), 130.0 (C-5), 108.8 (C-6), 156.8 (C-7), 104.0 (C-8), 115.9 (C-9), 154.6 (C-10), 128.4 (C-1’), 147.2 (C-2’), 141.1 (C-3’), 152.1 (C-4’), 121.8 (C-5’), 103.2 (C-6’), 60.5 (C-3’-O*C*H_3_), 55.8 (C-4’-O*C*H_3_), 100.8 (C-1’’), 73.7 (C-2’’), 76.6 (C-3’’), 69.5 (C-4’’), 77.1 (C-5’’), 60.8 (C-6’’) were carbon signal of astraisoflavan-7-*O*-glucoside [36]. And 100.9 (C-1’’’), 73.3 (C-2’’’), 75.8 (C-3’’’), 71.5 (C-4’’’), 75.7 (C-5’’’), 170.1 (C-6’’’) were carbon signal of glucuronide. According to HMBC spectra, δ4.90, which was the terminal hydrogen signal of glucuronide, related to the δ147.2 (C-2’), which indicated that glucuronide was linked to C-2’. At the same time, the terminal hydrogen of glucoside δ4.79 was in correlated with glucoside carbon at δ156.8 (C-7), which indicated that glucoside was linked to C-7. Considering all of the above, **MI-4** (**M106**) was determined as astraisoflavan-7-*O*-glucoside-2’-*O*-glucuronide (Figure 3d). It was a compound that isolated and identified by NMR for the first time. And its NMR spectroscopy was shown in Appendix A.

### 2.4. ARTF-Related Pharmacological Effect of Compounds In Vivo

The pharmacological literature of over 40 existence forms of ARTF which have specific or potential structure were retrieved from SciFinder and then analyzed. We found that 13 existence forms showed related pharmacological effect to ARTF, such as cardiovascular protective, neuroprotective, anti-inflammatory, and so on (Appendix A). Six of them were original constituents, namely calycosin (**F1**), calycosin-7-*O*-glucoside (**F4**), ononin (**F5**), formononetin (**F6**), daidzein (**F11**), naringin (**F18**); seven of them were metabolites, namely daidzein-4’-*O*-sulfate (**M107**), daidzein-7-*O*-sulfate (**M108**), daidzein-7-*O*-glucuronide or daidzein-4’-*O*-glucuronide (**M110**), genistein (**M118**), genistein-7-*O*-sulfate and genistein-4′-*O*-sulfate (two of **M119–M121**), and equol-7-*O*-sulfate (**M132** or **M133**). From Appendix A, we could find that the metabolites in Appendix A, especially phase Ⅱ metabolites, could activate estrogen receptor (ER). ER activation is associated with cardiovascular protective [37] and anti-inflammatory [38] effects, which is the main pharmacological effect of ARTF. In addition, we predicted that sulfate of flavonoids might have an effect on ER by molecular docking technique (data not shown). Furthermore, phase Ⅱ metabolites, especially sulfates, were the main existence forms of ARTF, so it could be speculated that some existence forms in vivo might be the material bases of the efficacies of ARTF, i.e., its effective forms.

## 3. Materials and Methods

### 3.1. Chemicals and Materials

ARTF (lot: 20170730) was obtained from Shanxi Baoji Herbest Biotech Co., Ltd. (Baoji, Shanxi, China) in August 2017 and its content was 62.7%, and the content of six main flavonoids namely calycosin-7-*O*-glucoside, calycosin, ononin, formononetin, astrapterocarpan-3-*O*-glucoside, and asisoflavan-7-*O*-glucoside was 11.3, 6.3, 5.8, 5.4, 7.2, and 1.6%, respectively, which was detected by HPLC-DAD-ELSD and calculated by using area normalization of ELSD (Evaporative Light Scattering Detector) chromatogram (Appendix A). The analysis of its constituents was conducted by HPLC-DAD-ESI-IT-TOF-MS^n^, and 69 constituents had been identified (Appendix A).

Calycosin (lot: MUST-16031110), calycosin-7-*O*-glucoside (lot: MUST-16031205) were brought from Chengdu Must Biotech Co., Ltd. (Chengdu, Sichuang, China). Formononetin (lot: LS60Q22) was supplied by Beijing J&K Scientific Co., Ltd. (Beijing, China). Astrapterocarpan (lot: PRF7042221), astrapterocarpan-3-*O*-glucoside (lot: PRF7043003), astraisoflavan (lot: PRF7120221) were provided by Chengdu Biopurify Phytochemicals Ltd. (Chengdu, Sichuang, China). Astraisoflavan-7-*O*-glucoside (lot: 160217) was purchased from Chengdu Pufei De Biotech Co., Ltd. (Chengdu, Sichuang, China). Ononin was isolated in our laboratory [36]. The purities for all reference compounds were over 98%.

Acetonitrile (HPLC grade, lot: 184866), formic acid (LC/MS grade, lot: 182088) were purchased from Thermo Fisher Scientific (Waltham, MA, USA). Methanol was supplied by Tianjing Damao Co., Ltd. (Tianjing, China). Sodium carboxymethyl cellulose (CMC-Na, Analytical grade) was purchased from Tianjing Guangfu Fine Chemical Research Institute (Tianjin, China). XAD-2 macroporous resins (lot: 94664), ODS (lot:9833), sephadex LH-20 were applied by Supelco (Bellefonte, PA, USA), YMC (Kyoto, Japan), and Amersham Biociences (Boston, MA, USA), respectively. Ultra-pure water was obtained by a Millipore Milli-Q Integral 3 Ultrapure water system (Billerica, MA, USA).

### 3.2. Animals and ARTF Administration

Fifty male Sprague-Dawley (SD) rats (250–300g) were obtained from the Experimental Animal Center of Peking University Health Science Center (Beijing, China) and 40 of them were kept in metabolic cages (Type: DXL-DL, Suzhou Fengshi Laboratory Animal Equipment Co. Ltd. (Suzhou, Jiangsu, China)) with two rats in each cage, and the other 10 rats were kept in normal cages, for free water and food twice a day. All the animals were maintained in an environmentally controlled breeding room for two days. Then, the next two days, urine and feces samples were collected twice a day and combined as blank samples. After that, the 40 rats in metabolic cages were orally administered with ARTF at 200 mg/kg (the suspension of ARTF at 25 mg/kg was prepared with 0.5% CMC-Na in an ultrasonic bath), twice a day (8:00 am and 8:00 pm) for 31 days, and the 10 rats in normal cages weren’t administered anything except for water and food. The animal experiments were approved by the Biomedical Ethical Committee of Peking University (approval No. LA2016205).

### 3.3. Bio-Samples Collection and Pre-Treatment

#### 3.3.1. Urine Collection and Pre-Treatment

Urine samples were collected twice a day (8:00 am and 8:00 pm) after administration of ARTF from metabolic cages and merged together, then filtered to remove impurities such as hair and dried in vacuum at 50 °C using a Heidolph Laborota 4001 rotatory evaporator (Heidolph Instruments GmhH & Co., Schwabach, Germany). After that, at a ratio of 1.0 g, dried samples were reconstituted in 10 mL methanol followed by 30 min ultrasonic extraction and filtered, then the filtrate was dried in vacuum (ca 25 g/day obtained) and 1.0 g sample was added 1 mL methanol to resuspended and stored at −20 °C. After 31 days, all the processed urine samples were mixed together as one sample and dried in vacuum. g). 1.0 g of the urine extract was taken out and redissolved in 5 mL methanol, then centrifuged at 15,000 rpm for 15 min and the supernatant was filtered through 0.45 μm nylon filter (Tianjin jinteng Experiment Co. Ltd., Tianjin, China). Finally, the filtrate was transferred into sample injection vial waiting for LC/MS analysis.

#### 3.3.2. Feces Collection and Pre-Treatment

Feces samples were collected twice a day (8:00 am and 8:00 pm) from metabolic cages. Feces samples of each day were dried in 50 °C for 48 h and were crushed into powder. A pulverized sample of 1.0 g was extracted with 5 mL methanol for 30 min in an ultrasonic bath three times. Afterward, the filtrates were combined and concentrated to dryness, then redissolved in 10 folds methanol to store at −20 °C. All feces samples were combined after processing, and concentrated to dryness as ARTF-containing feces extract; 0.3 g was taken out to resuspended in 3 mL methanol and centrifuged at 15,000 rpm for 15 min, and filtered through 0.45 μm nylon filter before LC/MS analysis.

#### 3.3.3. Plasma Collection and Pre-Treatment

At 32nd day, blood samples from rats in metabolic cages were collected by heart puncture technique under anesthesia at 10 min, 30 min, 1 h, 2 h, and 4 h (eight rats were sacrificed at each time point), and centrifuged at 5000 rpm for 15 min to obtain ARTF-containing plasma. Blank plasma was collected and processed in the same way from the rats in normal cages. All ARTF-containing and blank plasma were merged respectively, and 10 folds volume of methanol was added, and ultrasonically vibrated for 30 min to precipitate protein. Then, the mixture was centrifuged at 5000 rpm for 15 min, and the supernatant was condensed and dissolved in methanol, and the volume of methanol was 1% of the initial volume of plasma, and then centrifuged at 15,000 rpm for 15 min, finally filtered through 0.45 μm nylon filter before LC/MS analysis.

#### 3.3.4. Organs Collection and Pre-Treatment

After blood collection, the heart, liver, spleen, lung, kidney, stomach, small intestine, colon, thymus, and brain were quickly removed and flushed clearly with stroke-physiological saline solution until there was no obvious blood or content in the surface or cavity. All the organs were stored at –80℃. All the organs were shredded and suspended in deionized water at a ratio of 1.0 g to 4 mL, then homogenized by ultrasound homogenizer (Ultra-Turrax T8, Ika-werke Gmbh & Co. KG, Staufen, Germany). After that, 8 mL homogenates were extracted with 10-folds volume methanol in an ultrasonic bath for 30 min. The mixture was centrifuged at 5000 rpm for 15 min, and the supernatant was separated, dried, and resuspended in 2 mL methanol and filtered through 0.45 μm nylon filter before LC/MS analysis.

### 3.4. Isolation and Identification of Metabolites from ARTF-Containing Urine

Isolation procedure: ARTF-containing urine extract obtained in Section 3.3 (ca. 750 g), was dissolved in 1.5 L deionized water, filtered, and then subjected to XAD-2 macroporous resins column chromatography. Water, 20% methanol-water, 60% methanol-water, and 100% methanol were used to elute the column and get Fraction 1 to Fraction 4, respectively. The four metabolites, **MI-1** (**M108**) (2.67 mg), **MI-2** (**M32**) (818.30 mg), **MI-3** (16.66 mg), **MI-4** (**M106**) (34.62 mg), were isolated and purified from Fraction-3 by ODS column chromatography, Sephadex LH-20 column chromatography, and a Shimadzu preparative HPLC system sequentially, and their purity was above 90% determined by an Agilent 1200 HPLC.

Structure identification using NMR: MI-1 and the other three metabolites were dissolved in 0.15 mL and 0.5 mL DMSO-d6, respectively. Their ^1^H, ^13^C, heteronuclear singular quantum correlation (HSQC), heteronuclear multiple bond correlation (HMBC). NMR spectra were recorded on a Bruker DRX-400 NMR spectrometer (Bruker, Rheinstetten, Germany), using tetramethylsilane (TMS) as internal standard. All chemical shifts were reported in parts per million (ppm, δ), and coupling constants (*J*) in Hertz. UV spectra (200–400 nm) and HRMS data were recorded on the LC-MS-IT-TOF instrument with a PDA detector.

### 3.5. Instruments and Conditions

HPLC-DAD-ESI-IT-TOF-MS^n^ analyses were performed on a Shimadzu HPLC instrument (two LC-20AD pumps, an SIL-20AC autosampler, a CTO-20A column oven, an SPD-M20A PDA detector, a CBM-20A system controller) coupled with an IT-TOF mass spectrometer (Shimadzu, Kyoto, Japan) through an ESI interface. All data were processed by Shimadzu software, specifically, LCMS solution Version 3.60, Formula Predictor Version 1.2. and Accurate Mass Calculator. The chromatography separations were performed on an Industries Epic C_18_ column (250mm × 4.6 mm, 5 μm) (New Brunswick, NJ, USA) protected with an Agilent ZORBAX SB C_18_ column (12.5 mm × 4.6 mm, 5 μm) (Santa Clara, CA, USA). The mobile phase consisted of water-formic acid (100:0.1, *v*/*v*) (A) and acetonitrile (B) at a flow rate of 10,000 mL/min. A gradient elution program was adopted, specifically as 5% B at 0–10 min, 5–16% B at 10–12 min, 16–20% B at 12–25 min, 20–22% B at 25–45 min, 22–35% B at 45–60 min, 35–60% B at 60–85 min, 60–100% at 85–90 min. At the end of each run, 100% B was used to flush the column for 20 min. For mass detection, the mass spectrometer was programmed to carry out a full scan over *m*/*z* 100–1500 (MS^1^) with 30 ms accumulation time and *m*/*z* 50–1500 (MS^2^ and MS^3^) with 20 ms accumulation in both positive ion (PI) and negative ion (NI) detection mode; the flow rate was 0.2000 mL/min; the heat block and curved desolvation line temperature was 200 °C; the nebulizing nitrogen gas flow was 1.5 L/min; the interface voltage was (+), 4.5 kV; (−), 3.5 kV; the detector voltage was 1.7 kV; the relative collision-induced dissociation energy was 50%.

## 4. Conclusions

In summary, 170 kinds of compound (23 original constituents and 147 metabolites) were identified after administration of ARTF to rats, which included three newly detected original constituents and 89 new metabolites of ARTF, and 12 were regarded as new compounds (they are all metabolites) by retrieving information from the Sci inder database. Nineteen original constituents and 65 metabolites were detected and characterized in 10 organs. Four metabolites, including a new compound (calycoisn-3’-*O*-glucuronide methyl ester) and a first-isolated compound (astraisoflavan-7-*O*-glucoside-2’-*O*-glucuronide), along with two known compounds (daidzein-7-*O*-sulfate and calycosin-3’-*O*-glucuronide) were isolated from ARTF-containing urine and identified by NMR. Although the bioactivity studies of phase Ⅱ metabolites were little, 13 compounds (six original constituents, one phase I metabolite, six phase Ⅱ metabolites) in vivo were reported to possess similar pharmacological effects with ARTF, which indicated that they were effective forms of ARTF, and phase II metabolites might contribute to the efficacies of ARTF in vivo.

In the future, firstly, more kinds of phase I and phase II metabolites of ARTF should be obtained by synthesis or biotransformation. Then, the bioactivities of these metabolites should be determined to clarify the effective forms of ARTF. After that, the action mechanism of the effective forms can be studied. Finally, a new strategy to evaluate and control the quality of AR can be established.

## Figures and Tables

**Figure 1 molecules-25-05560-f001:**
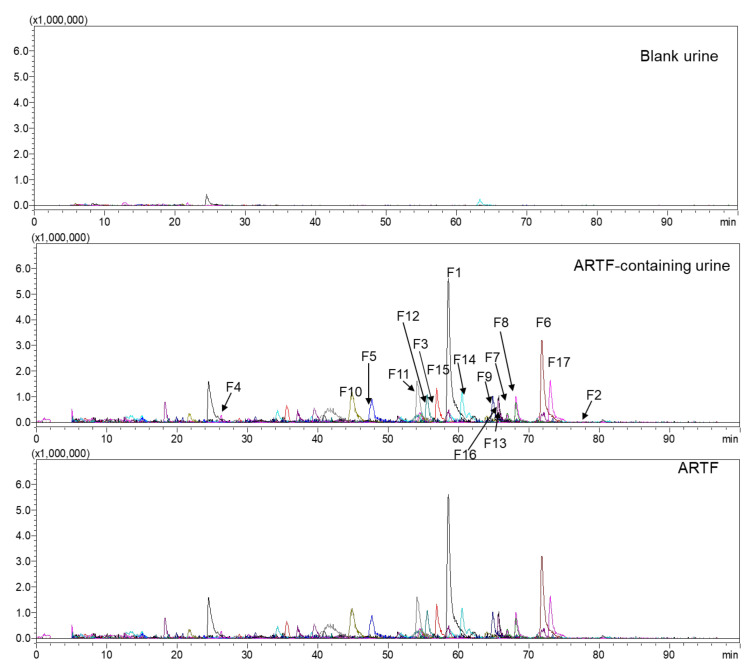
The extracted ion chromatograms (EICs) of original constituents (**F1**–**F17**) in rat urine after administration of Astragali Radix total flavonoid (ARTF).

**Figure 2 molecules-25-05560-f002:**
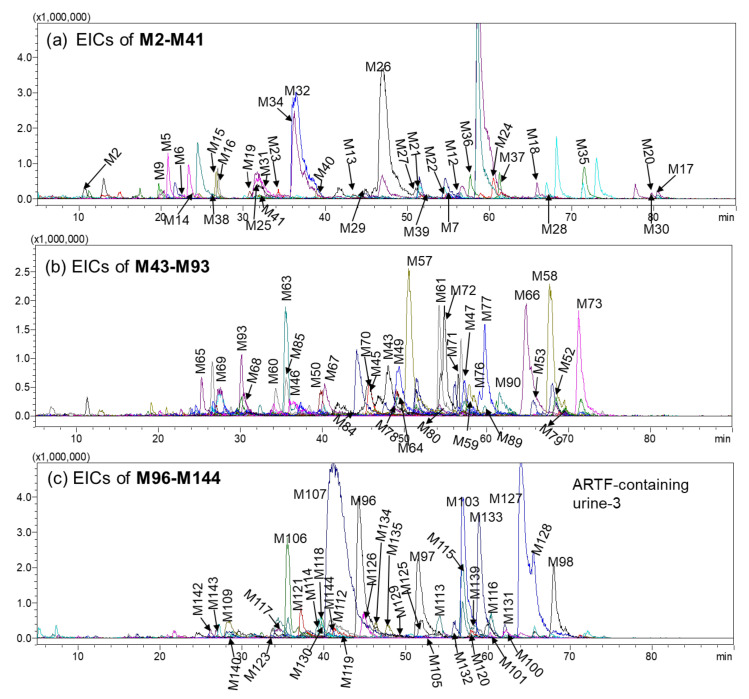
The EICs of 106 metabolites in rat urine after administration of ARTF. (**a**) EICs of **M2**–**M41**; (**b**) EICs of **M43**–**M93**; (**c**) EICs of **M96**–**M144**.

**Figure 3 molecules-25-05560-f003:**
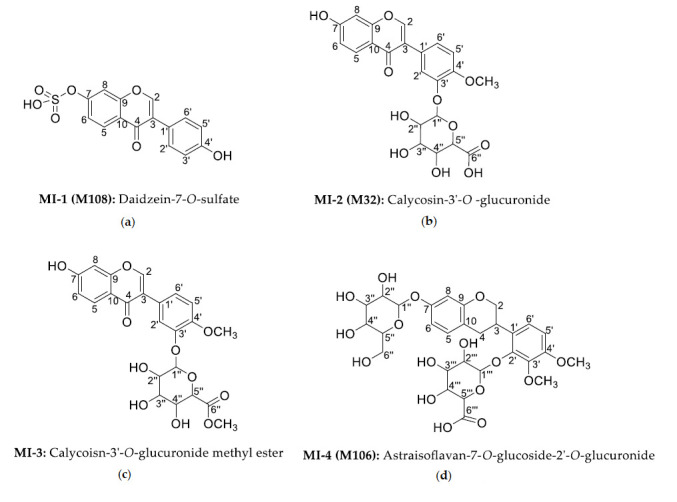
The structures of isolated metabolites from drug-containing urine. (**a**) **MI-1** (**M108)**; (**b**) **MI-2** (**M32)**; (**c**) **MI-3**; (**d**) **MI-4** (**M106)**.

**Table 1 molecules-25-05560-t001:** Original constituents in vivo after administration of ARTF to rats.

No.	t_R_ (min)	Formula (M)	Ion	Meas. (*m/z*)	Pred. (*m/z*)	Diff (ppm)	DBE	Identification	Plasma	Urine	Feces
F1 ^♥^	58.602	C_16_H_12_O_5_	[M − H]^−^	283.0623	283.0612	3.89	11	Calycosin		**▲**	▲
F2	77.588	C_16_H_12_O_5_	[M + H]^+^	285.0753	285.0758	−1.75	11	Calycosin isomer 1		▲	
F3 *	56.485	C_16_H_12_O_5_	[M − H]^−^	283.0608	283.0612	−1.41	11	Calycosin isomer 2		▲	
F4 ^♥^	27.200	C_22_H_22_O_10_	[M + HCOO]^−^	491.1215	491.1195	4.07	12	Calycosin-7-*O*-glucoside	▲	▲	
F5 ^♥^	47.743	C_22_H_22_O_9_	[M + H]^+^	431.1322	431.1337	−3.48	12	Ononin	▲	▲	
F6 ^♥^	71.750	C_16_H_12_O_4_	[M − H]^−^	267.0673	267.0663	3.74	11	Formononetin	▲	▲	▲
F7 *	66.802	C_16_H_12_O_6_	[M − H]^−^	299.0569	299.0561	2.68	11	Pratensein/Rhamnocitrin/5,7,4’-trihydroxy-3’-methoxyisoflavone		▲	
F8 *	68.197	C_16_H_12_O_6_	[M − H]^−^	299.0576	299.0561	5.02	11	Pratensein/Rhamnocitrin/5,7,4’-trihydroxy-3’-methoxyisoflavone		▲	
F9	65.018	C_15_H_12_O_5_	[M − H]^−^	271.0622	271.0612	3.69	10	Trihydroxy-dihydroisoflavone/flavone		▲	
F10	44.750	C_15_H_14_O_5_	[M − H]^−^	273.0777	273.0768	3.30	9	Trihydroxy-tetrahydroisoflavone/flavone		▲	
F11	54.377	C_15_H_10_O_4_	[M − H]^−^	253.0514	253.0506	3.16	11	Daidzein		▲	
F12	55.485	C_15_H_12_O_4_	[M − H]^−^	255.0675	255.0663	4.70	10	Trihydroxychalcone		▲	▲
F13	65.627	C_15_H_10_O_5_	[M − H]^−^	269.0452	269.0455	−1.12	11	Trihydroxyisoflavone/flavone		▲	
F14	60.827	C_17_H_14_O_6_	[M + H]^+^	315.0882	315.0863	6.03	11	Odoratin isomer		▲	
F15	57.460	C_17_H_18_O_5_	[M + H]^+^	303.1216	303.1227	−3.63	9	Astraisoflavane isomer		▲	
F16	65.852	C_17_H_16_O_6_	[M + H]^+^	317.1040	317.1020	6.31	10	Dihydroxy dimethoxyisoflavone/flavone		▲	
F17 ^♥^	73.328	C_17_H_16_O_5_	[M + H]^+^	301.1052	301.1071	−6.31	10	Astraptercarpan		▲	▲
F18	35.440	C_27_H_32_O_14_	[M − H]^−^	579.1748	579.1719	5.01	12	Naringin			
F19	34.635	C_28_H_36_O_13_	[M − H]^−^	579.2103	579.2083	3.45	11	Dihydrocalycosin pentose glucoside			
F20	51.837	C_28_H_34_O_14_	[M − H]^−^	593.1899	593.1876	3.88	12	Astrapterocarpan pentose glucoside			
F21	63.302	C_16_H_14_O_5_	[M + H]^+^	287.0927	287.0914	4.53	10	3,10-dihydroxy-9-methoxypterocarpan			
F22	60.812	C_22_H_22_O_11_	[M − H]^−^	461.1114	461.1089	5.42	12	Pratensein glucoside/5,7’,4’-trihydroxy-3’-methoxyisoflavone glucoside			
F23	22.332	C_22_H_22_O_11_	[M − H]^−^	461.1112	461.1089	4.99	12	Pratensein glucoside/5,7’,4’-trihydroxy-3’-methoxyisoflavone glucoside			
Sum									3	17	4

Note: t_R_: Retention time; Meas.: measured; Pred.: predicted; Diff: difference; DBE: double bone equivalents. ^♥^ These constituents were identified by comparison with reference compounds; * New original constituents found in vivo after administration of ARTF. ▲ Detected.

**Table 2 molecules-25-05560-t002:** Metabolites in vivo after administration of ARTF to rats.

NO.	t*_R_*(min)	Formula (M)	Ion	Meas. (*m*/*z*)	Pred. (*m*/*z*)	Diff (ppm)	DBE	Identification	Urine	Plasma	Feces
M1 ^△^	19.500	C_7_H_6_O_5_S	[M − H]^−^	200.9862	200.9863	−0.50	5	Hydroxyl-benzaladehyde sulfate or isomer		▲	
M2 ^△^	10.948	C_7_H_8_O_6_S	[M − H]^−^	218.9967	218.9969	−0.91	4	Methyl pyrogallol sulfate or isomer	▲		
M3 ^△^	20.550	C_8_H_8_O_6_S	[M − H]^−^	230.9961	230.9969	−3.46	5	Hydroxyphenylacetic acid sulfate		▲	
M4 ^△^	28.350	C_8_H_8_O_6_S	[M − H^−^	230.9963	230.9969	−2.6	5	Hydroxyphenylacetic acid sulfate isomer		▲	
M5 ^△^	21.008	C_9_H_8_O_6_S	[M − H]^−^	242.9978	242.9969	3.7	6	Hydroxycinnamic acid sulfate 1	▲	▲	
M6 ^△^	23.533	C_9_H_8_O_6_S	[M − H]^−^	242.9960	242.9969	−3.70	6	Hydroxycinnamic acid sulfate 2	▲		
M7 ^△^	55.018	C_10_H_12_O_5_S	[M − H]^−^	243.0345	243.0333	4.94	5	Eugenol sulfate	▲		
M8 ^△^	36.100	C_8_H_8_O_7_S	[M − H]^−^	246.9921	246.9918	1.21	5	Vanillic acid sulfate		▲	
M9 ^△^	19.617	C_9_H_12_O_6_S	[M − H]^−^	247.0281	247.0282	−0.40	4	Homovanillyl alcohol sulfate	▲		
M10 ^△^	20.383	C_10_H_10_O_7_S	[M − H]^−^	273.0080	273.0074	2.20	6	Ferulic Acid sulfate		▲	
M11 ^△^	58.818	C_11_H_14_O_6_S	[M − H]^−^	273.0436	273.0438	−0.73	5	Methoxyeugenol sulfate 1		▲	
M12 ^△^	55.593	C_11_H_14_O_6_S	[M − H]^−^	273.0433	273.0438	−1.83	5	Methoxyeugenol sulfate 2	▲	▲	
M13 ^△^	43.617	C_11_H_12_O_7_S	[M − H]^−^	287.0237	287.0231	2.09	6	C_11_H_12_O_4_ sulfate	▲		
M14 ^△^	23.750	C_11_H_14_O_7_S	[M − H]^−^	289.0393	289.0387	2.08	5	Ethylhomovanillic acid sulfate 1	▲	▲	
M15 ^△^	26.642	C_11_H_14_O_7_S	[M − H]^−^	289.0397	289.0387	3.46	5	Ethylhomovanillic acid sulfate 2	▲	▲	
M16 ^△^	27.558	C_11_H_12_O_8_ S	[M − H]^−^	303.0173	303.018	−2.31	6	3’,5’-dimethoxy-4’-hydroxycinnamic acid sulfate	▲	▲	
M17	80.230	C_16_H_12_O_5_	[M − H]^−^	283.0604	283.0612	−2.83	11	Calycosin isomer 1	▲		
M18 ^△^	65.833	C_16_H_12_O_5_	[M + H]^+^	285.0767	285.0758	3.16	11	Calycosin isomer 2	▲		
M19 ^△^	31.508	C_16_H_12_O_5_	[M + H]^+^	285.0738	285.0758	−7.02	11	Calycosin isomer 3	▲	▲	
M20 ^△^	79.592	C_16_H_12_O_5_	[M + H]^+^	285.0777	285.0758	6.66	11	Calycosin isomer 4	▲		
M21	51.610	C_16_H_12_O_6_	[M + H]^+^	301.0688	301.0707	−6.31	11	Hydroxycalycosin	▲		
M22	54.902	C_17_H_14_O_6_	[M + H]^+^	315.0883	315.0863	6.35	11	Methoxycalycosin 1	▲		
M23	34.517	C_17_H_14_O_6_	[M + H]^+^	315.0836	315.0863	−8.57	11	Methoxycalycosin 2	▲		
M24	61.477	C_17_H_14_O_6_	[M + H]^+^	315.0840	315.0863	−7.3	11	Methoxycalycosin 3	▲		
M25	31.483	C_16_H_12_O_8_S	[M − H]^-^	363.0178	363.0180	−0.55	11	Calycosin sulfate 1	▲	▲	
M26	47.512	C_16_H_12_O_8_S	[M − H]^−^	363.018	363.0180	0	11	Calycosin sulfate 2	▲	▲	
M27	51.168	C_16_H_12_O_8_S	[M − H]^−^	363.0187	363.0180	1.93	11	Calycosin sulfate isomer 1	▲	▲	
M28 ^△^	67.248	C_16_H_12_O_8_S	[M − H]^−^	363.0174	363.0180	−1.65	11	Calycosin sulfate isomer 2	▲		
M29 ^△^	44.867	C_16_H_12_O_8_S	[M − H]^−^	363.0189	363.0180	2.48	11	Calycosin sulfate isomer 3	▲	▲	
M30 ^△^	79.697	C_16_H_12_O_8_S	[M − H]^−^	363.0193	363.0180	3.58	11	Calycosin sulfate isomer 4	▲		
M31 ^△^	33.158	C_16_H_12_O_11_S_2_	[M − H]^−^	442.9767	442.9748	4.29	11	Calycosin-7,3’-*O*-disulfate	▲	▲	
M32 ^△^	36.133	C_22_H_20_O_11_	[M − H]^−^	459.0952	459.0933	4.14	13	Calycosin-3’-*O*-glucuronide	▲	▲	
M33 ^△^	23.417	C_22_H_20_O_14_S	[M − H]^−^	539.0538	539.0501	6.86	13	Calycosin glucuronide sulfate 1		▲	
M34 ^△^	31.245	C_22_H_20_O_14_S	[M − H]^−^	539.0519	539.0501	3.34	13	Calycosin glucuronide sulfate 2	▲	▲	
M35	71.298	C_16_H_12_O_9_S	[M − H]^−^	379.0137	379.0129	2.11	11	Hydroxycalycosin sulfate 1	▲	▲	
M36	57.743	C_16_H_12_O_9_S	[M − H]^-^	379.0153	379.0129	6.33	11	Hydroxycalycosin sulfate 2	▲		
M37	61.243	C_16_H_12_O_9_S	[M − H]^−^	379.0151	379.0129	5.80	11	Hydroxycalycosin sulfate 3	▲		
M38 ^△^	26.125	C_22_H_20_O_12_	[M − H]^−^	475.0847	475.0882	−7.37	13	Hydroxycalycosin glucuronide 1	▲		
M39 ^△^	51.610	C_22_H_20_O_12_	[M − H]^−^	475.0918	475.0882	7.58	13	Hydroxycalycosin glucuronide 2	▲		
M40 ^△^	39.042	C_23_H_22_O_12_	[M − H]^−^	489.1063	489.1038	5.11	13	Methoxycalycosin glucuronide	▲	▲	
M41 ^△^	33.123	C_23_H_22_O_15_S	[M − H]^−^	569.0634	569.0607	4.74	13	Methoxycalycosin glucuronide sulfate	▲	▲	
M42 ^△^	20.108	C_28_H_30_O_16_	[M − H]^−^	621.1451	621.1461	−1.61	14	Calycosin-7-*O*-glucoside-3’-*O*-glucuronide		▲	
M43	48.258	C_16_H_14_O_8_S	[M − H]^−^	365.0341	365.0337	1.1	10	Dihydrocalycosin sulfate 1	▲		
M44	43.285	C_16_H_14_O_8_S	[M − H]^−^	365.0346	365.0337	2.47	10	Dihydrocalycosin sulfate 2		▲	
M45	45.652	C_16_H_14_O_8_S	[M − H]^−^	365.0356	365.0337	5.2	10	Dihydrocalycosin sulfate 3	▲	▲	
M46 ^△^	38.098	C_22_H_22_O_11_	[M − H]^−^	461.1111	461.1089	4.77	12	Dihydrocalycosin glucuronide	▲		
M47	57.285	C_16_H_14_O_9_S	[M − H]^−^	381.0282	381.0286	−1.05	10	Hydroxy dihydrocalycosin sulfate 1	▲	▲	
M48	93.980	C_16_H_14_O_9_S	[M − H]^−^	381.0296	381.0286	2.62	10	Hydroxy dihydrocalycosin sulfate 2	▲		
M49	49.422	C_16_H_14_O_9_S	[M − H]^−^	381.0300	381.0286	3.67	10	Hydroxy dihydrocalycosin sulfate 3	▲	▲	
M50	39.792	C_18_H_18_O_8_S	[M − H]^−^	393.0665	393.065	3.82	10	Dimethyl dihydrocalycosin sulfate	▲		
M51 ^△^	52.118	C_17_H_16_O_9_S	[M − H]^−^	395.0434	395.0442	−2.03	10	Methoxy dihydrocalycosin sulfate 1		▲	▲
M52 ^△^	68.430	C_17_H_16_O_9_S	[M − H]^−^	395.0457	395.0442	3.80	10	Methoxy dihydrocalycosin sulfate 2	▲		
M53	66.472	C_16_H_14_O_10_S	[M − H]^−^	397.0251	397.0235	4.03	10	Dihydroxyl dihydrocalycosin sulfate	▲		
M54	25.252	C_16_H_16_O_8_S	[M − H]^−^	367.0477	367.0493	−4.36	9	Tetrahydrocalycosin sulfate 1			▲
M55	19.342	C_16_H_16_O_8_S	[M − H]^−^	367.0496	367.0493	0.82	9	Tetrahydrocalycosin sulfate 2			▲
M56	21.443	C_16_H_16_O_8_S	[M − H]^−^	367.0500	367.0493	1.91	9	Tetrahydrocalycosin sulfate 3			▲
M57	50.593	C_16_H_16_O_8_S	[M − H]^−^	367.0516	367.0493	6.27	9	Tetrahydrocalycosin sulfate 4	▲	▲	
M58	67.630	C_16_H_16_O_8_S	[M − H]^−^	367.0517	367.0493	6.54	9	Tetrahydrocalycosin sulfate 5	▲	▲	
M59	57.860	C_16_H_16_O_8_S	[M − H]^−^	367.0518	367.0493	6.81	9	Tetrahydrocalycosin sulfate 6	▲		
M60 ^△^	34.182	C_22_H_24_O_11_	[M − H]^−^	463.1257	463.1246	2.38	11	Tetrahydrocalycosin glucuronide 1	▲	▲	
M61 ^△^	54.027	C_22_H_24_O_11_	[M − H]^−^	463.1269	463.1246	4.97	11	Tetrahydrocalycosin glucuronide 2	▲	▲	
M62 ^△,^^★^	26.975	C_22_H_24_O_14_S	[M − H]^−^	543.0792	543.0814	−4.05	11	Tetrahydrocalycosin glucuronide sulfate		▲	
M63 ^△,^^★^	32.350	C_28_H_32_O_17_	[M − H]^−^	639.1607	639.1567	6.26	13	Tetrahydrocalycosin diglucuronide	▲		
M64	48.375	C_16_H_16_O_9_S	[M − H]^−^	383.0449	383.0442	1.83	9	Hydroxy tetrahydrocalycosin sulfate 1	▲		
M65	25.200	C_16_H_16_O_9_S	[M − H]^−^	383.046	383.0442	4.70	9	Hydroxy tetrahydrocalycosin sulfate 2	▲		▲
M66	64.843	C_16_H_16_O_9_S	[M − H]^−^	383.0462	383.0442	5.22	9	Hydroxy tetrahydrocalycosin sulfate 3	▲		
M67	40.192	C_16_H_16_O_9_S	[M − H]^−^	383.0467	383.0442	6.53	9	Hydroxy tetrahydrocalycosin sulfate 4	▲		
M68	30.608	C_16_H_16_O_9_S	[M − H]^−^	383.0444	383.0442	0.52	9	Hydroxy tetrahydrocalycosin sulfate 5	▲		
M69 ^△,^^★^	27.617	C_16_H_16_O_12_S_2_	[M − H]^−^	463.0031	463.0010	4.75	9	Hydroxy tetrahydrocalycosin disulfate	▲		
M70 ^△,^^★^	45.733	C_18_H_20_O_10_S	[M − H]^−^	427.0716	427.0704	2.81	9	Dihydroxy dihydrocalycosin sulfate	▲		
M71 ^△^	56.593	C_24_H_28_O_12_	[M − H]^−^	507.1535	507.1508	5.32	11	Dimethyl hydroxy tetrahydrocalycosin glucuronide 1	▲	▲	
M72 ^△^	54.727	C_24_H_28_O_12_	[M − H]^−^	507.1539	507.1508	6.11	11	Dimethyl hydroxy tetrahydrocalycosin glucuronide 2	▲	▲	
M73 ^△^	71.300	C_16_H_12_O_7_S	[M − H]^−^	347.0228	347.0231	−0.86	11	Formononetin-7-*O*-sulfate	▲	▲	
M74 ^△^	49.357	C_22_H_20_O_10_	[M − H]^-^	443.0977	443.0984	−1.58	13	Formononetin-7-*O*-glucuronide		▲	
M75	21.960	C_16_H_16_O_7_S	[M − H]^−^	351.0537	351.0544	−1.99	9	Tetrahydroformononetin sulfate 1			▲
M76	59.352	C_16_H_16_O_7_S	[M − H]^−^	351.0554	351.0544	2.85	9	Tetrahydroformononetin sulfate 2	▲	▲	
M77	59.860	C_16_H_16_O_7_S	[M − H]^−^	351.0564	351.0544	5.70	9	Tetrahydroformononetin sulfate 3	▲		
M78 ^△^	49.257	C_22_H_24_O_10_	[M − H]^−^	447.1322	447.1297	5.59	11	Tetrahydroformononetin glucuronide	▲	▲	
M79 ^△^	69.797	C_17_H_16_O_8_S	[M − H]^−^	379.0509	379.0493	4.22	10	Astrapterocarpan-3-*O*-sulfate	▲		
M80 ^△^	56.043	C_23_H_24_O_11_	[M − H]^−^	475.1276	475.1246	6.31	12	Astrapterocarpan-3-*O*-glucuronide	▲	▲	
M81	31.018	C_18_H_18_O_6_	[M − H]^−^	329.1027	329.1031	−1.22	10	Methoxyastrapterocarpan			▲
M82 ^△^	53.593	C_23_H_24_O_12_	[M − H]^−^	491.1225	491.1195	6.11	12	Hydroxyastrapterocarpan glucuronide 1		▲	
M83 ^△^	40.915	C_23_H_24_O_12_	[M − H]^−^	491.1194	491.1195	−0.20	12	Hydroxyastrapterocarpan glucuronide 2		▲	
M84 ^△^	43.618	C_23_H_24_O_12_	[M − H]^−^	491.1173	491.1195	−4.48	12	Hydroxyastrapterocarpan glucuronide 3	▲		
M85 ^△^	35.667	C_17_H_18_O_5_	[M +H]^+^	303.1205	303.1227	−7.26	9	Astraisoflavan isomer	▲		
M86 ^△^	32.868	C_18_H_20_O_5_	[M − H]^−^	315.1228	315.1238	−3.17	9	Methoxyastraisoflavan			▲
M87 ^△^	37.802	C_18_H_20_O_5_	[M − H]^−^	315.1235	315.1238	−0.95	9	Methoxyastraisoflavan isomer			▲
M88 ^△^	34.368	C_19_H_22_O_6_	[M − H]^-^	345.1360	345.1344	4.64	9	Hydroxy dimethoxyastraisoflavan			▲
M89	59.918	C_17_H_18_O_8_S	[M − H]^−^	381.0639	381.0650	−2.89	9	Astraisoflavan-7*-O*-sulfate	▲		
M90	62.510	C_17_H_18_O_8_S	[M − H]^−^	381.0660	381.0650	2.62	9	Astraisoflavan-2’-*O*-sulfate	▲	▲	
M91	34.427	C_17_H_18_O_8_S	[M − H]^−^	381.0662	381.0650	3.15	9	Astraisoflavan sulfate isomer			▲
M92 ^△^	34.993	C_18_H_20_O_8_S	[M − H]^−^	395.0795	395.0806	−2.78	9	Methyoxyastraisoflavan sulfate 1			▲
M93 ^△^	30.460	C_18_H_20_O_8_S	[M − H]^−^	395.0819	395.0806	3.29	9	Methyoxyastraisoflavan sulfate 2	▲		▲
M94	20.252	C_17_H_18_O_9_S	[M − H]^−^	397.0602	397.0599	0.76	9	Hydroxyastraisoflavan sulfate 1			▲
M95	49.543	C_17_H_18_O_9_S	[M − H]^−^	397.0603	397.0599	1.01	9	Hydroxyastraisoflavan sulfate 2			▲
M96	44.008	C_17_H_18_O_9_S	[M − H]^−^	397.0608	397.0599	2.27	9	Hydroxyastraisoflavan sulfate 3	▲	▲	
M97 ^△^	51.435	C_17_H_18_O_9_S	[M − H]^−^	397.0614	397.0599	3.78	9	Hydroxyastraisoflavan sulfate 4	▲		
M98 ^△^	68.372	C_17_H_18_O_9_S	[M − H]^−^	397.0620	397.0599	5.29	9	Hydroxyastraisoflavan sulfate 5	▲		
M99 ^△^	18.817	C_17_H_18_O_9_S	[M − H]^−^	397.0622	397.0599	5.79	9	Hydroxyastraisoflavan sulfate 6			▲
M100 ^△^	62.393	C_18_H_20_O_9_S	[M − H]^−^	411.0743	411.0755	−2.92	9	Methoxyastraisoflavan sulfate 1	▲		
M101 ^△^	60.210	C_18_H_20_O_9_S	[M − H]^−^	411.0758	411.0755	0.73	9	Methoxyastraisoflavan sulfate 2	▲		
M102 ^△^	58.702	C_23_H_26_O_11_	[M − H]^−^	477.1429	477.1402	5.66	11	Astraisoflavan-7-*O-*glucuronide		▲	
M103 ^△^	56.868	C_23_H_26_O_11_	[M − H]^−^	477.1430	477.1402	5.87	11	Astraisoflavan-2’-*O*-glucuronide	▲	▲	
M104 ^△,^^★^	38.250	C_23_H_26_O_14_S	[M − H]^−^	557.0997	557.0970	4.85	11	Astraisoflavan glucuronide sulfate 1		▲	
M105 ^△,^^★^	52.510	C_23_H_26_O_14_S	[M − H]^−^	557.1001	557.0970	5.56	11	Astraisoflavan glucuronide sulfate 2	▲		
M106 ^△^	35.633	C_29_H_36_O_16_	[M − H]^−^	639.1949	639.1931	2.82	12	Astraisoflavan-7-*O*-glucoside-2’-*O*-glucuronide	▲	▲	
M107	41.557	C_15_H_10_O_7_S	[M − H]^−^	333.0074	333.0074	0	11	Daidzein-4’-*O*-sulfate	▲	▲	
M108	28.925	C_15_H_10_O_7_S	[M − H]^−^	333.0080	333.0074	1.80	11	Daidzein-7-*O*-sulfate		▲	
M109 ^△^	28.400	C_15_H_10_O_10_S_2_	[M − H]^−^	412.9658	412.9643	3.63	11	Daidzein-7,4’-*O*-disulfate	▲	▲	
M110 ^△^	23.817	C_21_H_18_O_10_	[M − H]^−^	429.0835	429.0827	1.86	13	Daidzein glucuronide		▲	
M111 ^△^	20.167	C_21_H_18_O_13_S	[M − H]^−^	509.0403	509.0395	1.57	13	Daidzein glucuronide sulfate		▲	
M112 ^△^	44.292	C_15_H_12_O_7_S	[M − H]^−^	335.0212	335.0231	−5.67	10	Dihydrodaidzein sulfate	▲		
M113	53.852	C_15_H_14_O_7_S	[M − H]^−^	337.0389	337.0387	0.59	9	Tetrahydrodaidzein sulfate 1	▲		
M114	39.333	C_15_H_14_O_7_S	[M − H]^−^	337.0400	337.0387	3.86	9	Tetrahydrodaidzein sulfate 2	▲		
M115	57.002	C_15_H_14_O_7_S	[M − H]^−^	337.0402	337.0387	4.45	9	Tetrahydrodaidzein sulfate 3	▲	▲	
M116	60.560	C_15_H_14_O_7_S	[M − H]^−^	337.0409	337.0387	6.53	9	Tetrahydrodaidzein sulfate 4	▲		
M117 ^△^	34.458	C_15_H_14_O_10_S_2_	[M − H]^−^	416.9988	416.9956	7.67	9	Tetrahydrodaidzein disulfate	▲		
M118	40.017	C_15_H_10_O_5_	[M + H]^+^	271.0617	271.0601	5.90	11	Gensitein	▲		
M119	41.833	C_15_H_10_O_8_S	[M − H]^−^	349.0033	349.0024	2.58	11	Genistein sulfate 1	▲		
M120	57.918	C_15_H_10_O_8_S	[M − H]^−^	349.0039	349.0024	4.30	11	Genistein sulfate 2	▲		
M121	37.233	C_15_H_10_O_8_S	[M − H]^−^	349.0030	349.0024	1.72	11	Genistein sulfate 3	▲		
M122 ^△^	25.042	C_21_H_18_O_14_S	[M − H]^−^	525.0367	525.0370	−0.57	9	Genistein glucuronide sulfate		▲	
M123	33.658	C_15_H_12_O_8_S	[M − H]^−^	351.0180	351.0180	0	10	Dihydrogenistein sulfate 1	▲		
M124 ^△^	74.098	C_15_H_12_O_8_S	[M − H]^−^	351.0184	351.0180	1.14	10	Dihydrogenistein sulfate 2		▲	
M125 ^△^	51.688	C_15_H_12_O_8_S	[M − H]^−^	351.0189	351.018	2.56	10	Dihydrogenistein sulfate 3	▲		
M126	44.758	C_15_H_14_O_5_	[M − H]^−^	273.0757	273.0768	−4.03	9	Tetrahydrogenistein	▲		
M127	63.985	C_15_H_14_O_8_S	[M − H]^−^	353.0359	353.0337	6.23	9	Tetrahydrogenistein sulfate 1	▲		▲
M128	65.410	C_15_H_14_O_8_S	[M − H]^−^	353.0360	353.0337	6.51	9	Tetrahydrogenistein sulfate 2	▲		
M129 ^△,^^★^	49.090	C_20_H_22_O_12_S	[M − H]^−^	485.0787	485.0759	5.77	10	Tetrahydrogenistein pentose sulfate	▲		
M130 ^△,^^★^	39.850	C_21_H_24_O_13_S	[M − H]^−^	515.0911	515.0865	8.93	10	Tetrahydrogenistin sulfate	▲		
M131	62.235	C_15_H_14_O_6_S	[M − H]^−^	321.0439	321.0438	0.31	9	Equol sulfate isomer	▲		
M132	56.152	C_15_H_14_O_6_S	[M − H]^−^	321.0444	321.0438	1.87	9	Equol sulfate 1	▲	▲	
M133	58.943	C_15_H_14_O_6_S	[M − H]^−^	321.0454	321.0438	4.98	9	Equol sulfate 2	▲		
M134 ^△^	46.017	C_21_H_22_O_9_	[M − H]^−^	417.1185	417.1191	−1.44	11	Equol glucuronide 1	▲	▲	
M135 ^△^	47.975	C_21_H_22_O_9_	[M − H]^−^	417.1209	417.1191	4.32	11	Equol glucuronide 2	▲	▲	
M136 ^△,^^★^	32.293	C_21_H_22_O_12_S	[M − H]^−^	497.0786	497.0759	5.43	11	Equol glucuronide sulfate 1		▲	
M137 ^△,^^★^	28.058	C_21_H_22_O_12_S	[M − H]^−^	497.0797	497.0759	7.64	11	Equol glucuronide sulfate 2		▲	
M138 ^△^	37.317	C_21_H_22_O_12_S	[M − H]^−^	497.0757	497.0759	-0.4	11	Equol glucuronide sulfate 3		▲	
M139 ^△^	57.977	C_15_H_16_O_6_S	[M − H]^−^	323.0616	323.0595	6.5	8	Dihydroequol sulfate	▲		
M140	28.817	C_16_H_14_O_6_	[M + H]^+^	303.0878	303.0863	4.95	10	Dihydropratensein	▲		
M141 ^△^	24.925	C_22_H_20_O_15_S	[M − H]^−^	555.0487	555.0450	6.67	13	Pratensein glucuronide sulfate		▲	
M142 ^△,^^★^	26.300	C_27_H_30_O_17_	[M − H]^−^	625.1462	625.1410	8.32	13	Tetrahydro trihydroxyisoflavone diglucuronide 1	▲		
M143 ^△,^^★^	26.867	C_27_H_30_O_17_	[M − H]^−^	625.1467	625.1410	9.12	13	Tetrahydro trihydroxyisoflavone diglucuronide 2	▲		
M144 ^△^	41.650	C_15_H_14_O_9_S	[M − H]^−^	369.0303	369.0286	4.61	9	Tetrahydro-tetrahydroxyisoflavone sulfate	▲		
M145 ^△^	41.930	C_22_H_20_O_15_S	[M − H]^−^	555.0478	555.0450	6.76	13	Pratensein glucuronide sulfate			
M146 ^△^	61.012	C_22_H_20_O_11_	[M − H]^−^	459.0925	459.0933	−1.23	13	Calycosin-7-*O*-glucuronide			
M147 ^△^	69.988	C_17_H_18_O_8_S	[M − H]^−^	381.0652	381.0650	0.52	9	Astraisoflavan sulfate isomer			
Sum									106	64	17

Note: t_R_: Retention time; Meas.: measured; Pred.: predicted; Diff: difference; DBE: double bone equivalents. ^△^ New metabolites found in vivo after administration of ARTF; ^★^ Potential New compound by retrieving information from SciFinder database. ▲ Detected.

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
