# Peer review of "Exploring the In Vivo Existence Forms (23 Original Constituents and 147 Metabolites) of Astragali Radix Total Flavonoids and Their Distributions in Rats Using HPLC-DAD-ESI-IT-TOF-MSn"

_molecules, 2020, doi:10.3390/molecules25235560_

Round 1

Reviewer 1 Report

The paper of Liu et al. presents a study on the biotransformation of a plant extract (a chinese traditional drug) from Astragali Radix (AR) in vivo in rats. The authors have analysed by HPLC/MS fractions from urine, plasma feces and tissue of rats after ingestion of this AR

The study is well made, and well constructed but suffers from probable lousy translation from chinese. Thus it is relatively difficult to read for a non English European. (I could manage, but had to read the sentences several times to make my mind)

The results reported are interesting. However the is a need for editing many sentences. The translator does not make the difference between past (existed) and present (existing) or the substantive (existence). However for the comprehension of the sentence it is important to use the correct term.

Thus I suggest the following corrections:

Page 1, line 22: existence forms change to : forms existing in vivo or compounds existing in vivo.

Page 1, line 32 change existed into : existing.

Page 2 line 55 : formononetin (15) were incubated

Page 2 line 65 : change existence form … into : molecular species found in vivo

Page 2 line 74 constituents and metabolites that existed in vivo

Page 3 line 101 : change not existed into : did not exist

Page 3, line 114, 115 : F1, F6, F17 (Figure S2) existed in the feces.

Page 3, Figure 1a. The figure is very difficult to read because of the m/z numbers on the left side of the chromatograms correlated to diverses colors for the line spectra.

This is the same for all chromatograms. Is it possible to make the lines thicker?

One other way would be to put the base peak (m/z) above each chromatogram peak.

Page 6 line 127: The peaks only appearing in LC-MS…

Page 6 line 132 : replace containing by : from which 12

Page 6, line 145 : Do not start the sentence by a numerical number. Thus not 138 but Hundred and thirty eight.

Page 6 line 150-151 as metabolites originating from flavone …, then conjugation with sulphuric…

Line 154 : I dispute the structure you give : I think you cannot make a sulphate of benzoic acid. It is probably an isomer perhaps hydroxyl-benzaldehyde sulphate, but other structures are possible.

Line 157 : methyl pyrogallol sulphate : other isomeric structures are possible. For instance 3-methoxy-catechol sulphate or any isomer.

There are less problems in the following pages.

Supplementary part.

Paragraph 3 : Fig S5-S16 : Add in the legend of the suplementary MS2 or MS3 spectra the value of the parent ion. It would be easier to read.

In the NMR spectra, on the 1H spectrum : Could you add the molecular structure of the molecule with the numbering as an insert ?

All together, the paper is of good quality, and interesting. It needs some editing because of the language imperfections.

The scientific part is convincing. Some attributed structures are probably incorrect or only tentative. This should be stated. The text is also quite compact. Perhaps adding some paragraphs would make it easier to read.

The authors would make their paper easier to read if they stated the flavonoid content of the AR extract given to rats. They could perhaps add one figure of a chromatogram of this extract. Because  hydroxylated flavonoids are still flavonoids and may exist in plant extracts.

However I think the paper only need minor revisions for the form and the language. Then it should be acceptable.

.

Author Response

Dear reviewer, thanks for your very good suggestions and English corrections for us. it's very helpful for us to improve the improve the quality of our paper. And the point-by-point response to your comment please see the attachment. Thanks a lot again.

Reviewer 2 Report

The work s very interesting and well performed. It describes the metabolic fate of Astragali Radix total flavonoids fraction in rats. Authors presented a lot of data, indicating known and new metabolites and original compounds in body fluids and organs of rats which were administered of ARTF for 31 days.

The procedures are correct but:

The determination of ARTF purity and chemical composition should be described or referenced (Lines 397-398).

The isolation of Ononin should be described or referenced (Line 406).

The procedure of isolation of compounds from urine should be described in detail, it could be placed in supplementary time.

Other suggestions

The abstract should be improved. It provides to many different numbers which are not easy to follow.  It should be more descriptive to provide the general outcome of the work, underlining the novelty and the significance of the study

Conclusions should provide some future directions, if possible suggestions what could be studied for bioactivity, which molecules are probable active ones. The data retained from SciFinder should also be more discussed.

Figure S17. – b) should be metabolites in liver!

Author Response

Dear reviewer, thanks as lot for your good suggestion for our manscript, and it help us much to improve tht quality for our paper. And the point-by-point response to your comments please see the attachment.

Reviewer 3 Report

The paper of Liu, et al. “Exploring the in vivo existence forms (23 original constituents and 147 metabolites) of Astragali Radix total flavonoids and their distributions in rats using HPLC-DAD-ESI-IT-TOF-MSn” aimed mainly to investigate the metabolites of Astragali Radix total flavonoids after in vivo administration. The overall work is well written (in my opinion) and has high scientific soundness. But I would like to make some little remarks that have arisen after reading the manuscript.

  1. Figure 1b is making me confusing a bit. Three chromatograms have the same caption as "Blank urine" and two more chromatograms described as "ARTF-containing urine". What are the differences between pictures with the same signs? Would you like to deal with this disadvantage?
  2. You have to use abbreviations and cutoff words in Table 1 correctly. I have no idea what "Meas.", "Pred.", "Diff", "DBE" should mean.
  3. How would you like to use "MDPI_2.3_heading3" to design some titles as "Identification of the sulfates of the ring cleavage products of flavone (M1-M16)", "Identification of the calycosin-related metabolites (M17-M72, M146)" etc?
  4. I suppose the lines 316-321, 327-333, 341-349, 360-367 with NMR description, may be easily replaced in the Material and Methods Section.
  5. The description of Astragali Radix total flavonoids (lines 396-397) as "...and its purity was 62.7%..." should be corrected. It's ok when you mean the purity of the separate compound. The content of the separate compounds in the ARTF should be mentioned

Finally, in my opinion, the paper needs a little improvement and maybe accepted after minor revision.

Author Response

Dear reviewer, thanks as lot for your good suggestion for our manscript, and it do much help for us to improve tht quality for our paper. And the point-by-point response to your comments please see the attachment.

Reviewer 4 Report

The manuscript represents a good analytical work in which authors have identified 170 kinds of compounds after administration of Astragali radix total flavonoids to rats. 

A proper experimental design was performed and the methods are appropriate. The aims of the paper is clearly defined and the discussion is supported by data. The conclusion is concise and objective.

The paper, according to my perspective deserves publication.

Author Response

Dear reviewer, thanks a lot for you reviewing and recognizing our manuscript. Thanks very much again.

Round 2

Reviewer 2 Report

The authors applied suggested changes hence the manuscript can be published in the present form